# The Demodifier: A tool for screening modification-induced alternate peptide taxonomy in palaeoproteomics

**Miranda Evans** [1,2,3]*

1 Department of Archaeology, University of Cambridge, Cambridge, United Kingdom, 2 Department of Zoology, University of Cambridge, Cambridge, United Kingdom, 3 BioArCh, Department of Archaeology, University of York, York, United Kingdom

* mae52@cam.ac.uk

## Abstract

In palaeoproteomic research, the accuracy of taxonomic matches is crucial, as research questions frequently hinge on which species were utilised by ancient people. However, protein modifications including deamidation of glutamine and asparagine, and conversion of glutamine or glutamic acid to pyroglutamic acid, can change the sequence of peptides, leading to spurious taxonomic detections and potentially inaccurate archaeological interpretations. While a handful of examples of this phenomenon have been reported in the literature, the issue is potentially much wider reaching than currently realised. In reality, any time a peptide containing a deamidated glutamine or asparagine residue, an unmodified glutamic acid or aspartic acid residue, or a pyroglutamic acid modification is detected by proteomic search software, the sequence, and therefore potentially its taxonomy, may be incorrect, which could potentially lead to unsound archaeological interpretations. The Demodifier is a fast, open source tool which solves this issue by screening for modification-induced alternate peptide taxonomy, enabling archaeologists to make informed interpretations of the taxonomies of peptides detected in ancient samples. To assess its utility, The Demodifier is tested against an archaeological dataset containing all unique peptides reported in palaeoproteomic studies of dental calculus and vessels. The results reveal that modification-induced alternate peptide taxonomies are severely underreported, occurring almost ten times more frequently than previously understood. Modifications were found to produce three different types of inaccurate taxonomic matches: those which yielded completely different taxonomic lowest common ancestors to the input peptide, those which were more taxonomically specific than the input peptide, and those which were less taxonomically specific than the input peptide. The Demodifier therefore enables the rapid detection of potentially inaccurate peptide taxonomies, avoiding spurious archaeological interpretations in future studies.

**Data availability statement:** The repository containing a static version of The Demodifier (1.4.0), used in the case study analysis is named "miranda-e/Demodifier: The Demodifier 1.4" and is available at https://doi.org/10.5281/zenodo.16961714. The updated version, The Demodifier (1.6.0), which is functionally the same but refactored for clarity and updated to enable additional input file types, following reviewer recommendation, is available under the repository name "miranda-e/Demodifier: v1.6.0" at https://doi.org/10.5281/zenodo.17689841. A tutorial for The Demodifier, and the dynamic version, are available in the Github repository named "Demodifier" at https://github.com/miranda-e/Demodifier. All other data are available within the article or its Supporting Information.

**Funding:** This work was funded by a Philip Leverhulme Prize award to Jessica Hendy and a UKRI Horizon Europe Guarantee (Grant No. EP/Y009878/1) awarded to Christiana Scheib. The funders had no role in study design, data collection and analysis, decision to publish, or preparation of the manuscript.

**Competing interests:** The authors have declared that no competing interests exist.

## Introduction and background

In palaeoproteomics, the reported taxonomy of ancient peptides is assessed by analysing the lowest common ancestor (LCA) of all the taxa to which a given peptide matches. It is crucial that these taxonomic assignments are secure, because they frequently form the basis of archaeological interpretations, e.g., the detection of newly introduced plant taxa [1], multitaxa dairying [2], or mixed ingredient processing [3]. Yet post-translational modifications (PTMs) including deamidation of glutamine (Q) and asparagine (N) and N-terminal pyroglutamic acid formation can alter peptide sequences, potentially leading to inaccurate taxonomic matches.

Deamidation is a PTM in which asparagine (N) is converted to aspartic acid (D), and glutamine (Q) is converted to glutamic acid (E) (Fig 1). This reaction involves the conversion of an amide group (-$CONH_2$) into a carboxylic acid (-COOH), resulting in a monoisotopic mass shift of +0.984 Da (3 d.p.), a change commonly observed in both peptide mass fingerprinting and tandem mass spectrometry analyses. Deamidation of both asparagine and glutamine can proceed via two pathways (Fig 1). At acidic pH, the direct hydrolysis pathway predominates, while under neutral to basic conditions, intramolecular cyclisation to a cyclic intermediate occurs (succinimide for asparagine, glutarimide for glutamine), although this pathway is less favourable for glutamine than for asparagine [4–6]. The cyclic intermediate pathway can generate isoform residues, isoaspartic acid and isoglutamic acid, which have identical masses to their non-isoform counterparts [6,7]. Asparagine (N) has been found to deamidate at a higher rate than glutamine (Q), which deamidates considerably more slowly [8]. Given that deamidation occurs at a relatively regular rate under constant conditions, it has been proposed as a molecular clock [8], a marker of preservational quality [9,10], or even an indicator of processing [11]. Deamidation rates have been used as a tool for establishing the authenticity of palaeontological, archaeological or heritage proteins, initially using bulk deamidation rates [10,12–15], and later using site-specific measures [16,17]. Deamidation rates are also impacted by protein structure [8,18,19], temperature [20], and pH [21,22].

Recognising that the mass of aspartic acid (D) is indistinguishable from deamidated asparagine (N), Warinner et al. [23] observed that such modification induced sequence changes could result in potentially incorrect LCA matches. In particular, they found that deamidation of an asparagine (N) residue in the *Ovis* whey protein β-lactoglobulin (βLG) peptide TPEVD**N**EALEK causes it to be indistinguishable from the Bovinae βLG peptide TPEVD**D**EALEK, meaning that the detection of TPEVD-**D**EALEK could equally be derived from Bovinae or *Ovis* dairy in ancient samples. Here I propose the term Modification Induced Sequence Permutation (MISP) to describe such peptide variants. Crucially in this case, the detection of inaccurate dairy species could lead to spurious interpretations of dairy exploitation and species introduction. Given that the peptide in question is more frequently detected than any other in ancient dental calculus and ceramic residues [3,24–26], this discovery has gained widespread application [e.g. 2,3,27,28].

Several similar MISPs have been noted in the interim, especially for other βLG peptides. Jersie-Christensen et al. [29] reported that the βLG peptide PTPEG**D**/

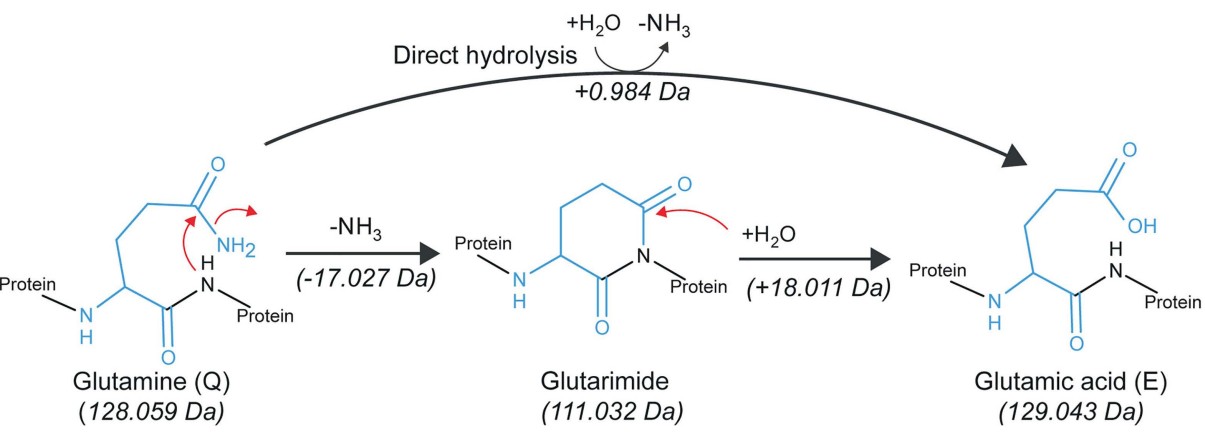

**Fig 1. Diagram showing deamidation reaction mechanisms. (A)** Deamidation of asparagine to aspartic acid. **(B)** Deamidation of glutamine to glutamic acid. Diagrams illustrate both direct hydrolysis and cyclic intermediate pathways. Note: the formation of isoaspartic acid and isoglutamic acid is not depicted.

**N**LEILLQK which they detected in medieval dental calculus samples could match to either Bovinae or *Ovis* respectively. Analysing βLG on neonate dog bones from 430–960 CE Japan, Tsutaya et al. [30] reported that the peptide TME**D/N**LD-LQK could belong to either Canidae, the most parsimonious interpretation, or alternatively to *Callorhinus ursinus* (northern fur seal), *Odobenus rosmarus divergens* (Pacific walrus), and *Leptonychotes weddellii* (Weddell seal). Lastly, in their analysis of a well-preserved Early Bronze Age wooden container from Switzerland, Colonese et al. [31] reported a MISP for the peptide TLFGS**D/N**PPIPTPVLTK, which could render its initial LCA match to *Hordeum vulgare* insecure. To date, no glutamine (Q) deamidation MISPs have been reported in the ancient protein literature, which is perhaps unsurprisingly given the lower rate of glutamine (Q) deamidation compared to asparagine (N), however, deamidation of glutamine could also potentially result in incorrect taxonomic detection. It is also possible that further examples have been recognised but

not reported in the literature, i.e., if researchers who do screen for MISPs simply provide additional LCAs in their results, without explaining that these were caused by modification-induced sequence changes.

Another PTM which causes amino acid mass shifts that could lead to MISPs and thus potentially inaccurate LCAs is the conversion of N-terminus glutamine or glutamic acid to pyroglutamic acid (Gln →pyroGlu or Glu →pyroGlu). Pyroglutamic acid forms via two different pathways (Fig 2). The loss of an amide group (i.e., deamidation) converts glutamine (Q) to pyroglutamic acid, or alternatively, the loss of a water molecule (i.e., dehydration) converts glutamic acid (E) to pyroglutamic acid. Pyroglutamic acid is formed through the intramolecular cyclisation of N-terminal glutamine (Q) or glutamic acid (E) residues via both non-enzymatic [32] and enzymatic processes [33,34]. This transformation can occur under physiological conditions or after an organism's death, and is influenced by factors such as temperature, pH, enzymatic activity, and pressure [35–38]. Due to the identical masses of glutamine (Q) minus $NH_3$, pyroglutamic acid, and glutamic acid (E) minus $H_2O$, any time a peptide with the Gln→pyroGlu (N-term Q) or Glu→pyroGlu (N-term E) modification is detected by search software, the peptide's N-terminus amino acid could equally be glutamic acid (E) or glutamine (Q). So far, the only instances of this PTM causing MISPs reported in the archaeological literature are in Evans [39]. However, its implications could potentially be far reaching.

Given the lack of software tools for MISP detection, researchers seeking to screen for MISPs are left to manually calculate peptide permutations and their taxonomies, which is a time-consuming task, and is yet to be standard practice. It is thus perhaps unsurprising that none of the reported examples involve multiple modification sites, despite the equal mass of multi-modification site MISPs, those with single modifications, and the original sequence. Peptides with numerous potential deamidation sites have drastically increased numbers of MISPs and therefore potentially taxonomies, making their detection time-consuming without programmatic tools. Despite the handful of reported examples, the issue of modification-induced changes to taxonomic detection is likely much more widespread than previously realised. In reality, any time a peptide containing a glutamic acid (E) or aspartic acid (D), a deamidated glutamine (Q) or deamidated asparagine (N), or a Gln/Glu→ pyroGlu modification is detected by the search software, the sequence and potentially its taxonomy may be incorrect.

To enable researchers to combat this issue, I present The Demodifier (1.6.0). This tool screens for all possible MISPs of a given list of peptides and their modifications, and assigns taxonomy using the Unipept API [40]. The aim of The Demodifier is to reveal peptides with MISPs which yield additional alternate taxonomies, enabling the researcher to more accurately interpret the taxonomies of peptides represented in their samples. I then use The Demodifier (1.4.0) to screen

## Pyroglutamic Acid Formation

Fig 2. **Diagram showing reaction mechanisms for conversion of N-terminal Glutamine and Glutamic Acid to Pyroglutamic Acid.** Note: formation of pyroglutamic acid from isoaspartic acid and isoglutamic acid is not depicted.

published ancient dietary peptides for alternative, equally possible taxonomic detections, with the aim of assessing its utility on a sample of real ancient peptides. The results reveal many MISPs with equifinal taxonomies, which have yet to be recognised in previous palaeoproteomic studies. As more ancient metaproteomic analyses are published, further modification-induced sequence permutations will likely be identified.

## Materials and methods

### The Demodifier

**Overview.** The Demodifier (1.6.0) screens for potential modification-induced sequence changes, which yield alternate taxonomies to the input peptide. It consists of a Python3 script which can be run either as an executable on Windows or Linux, or on the command line: https://doi.org/10.5281/zenodo.17689841. A tutorial for The Demodifier is available at https://github.com/miranda-e/Demodifier. The Demodifier requires the external libraries: requests (2.28.1) and urllib3 (1.26.7) and was written in Python 3.9.7. Fig 3 provides a schematic overview of The Demodifier's process. Briefly, The Demodifier accepts as input a CSV or TSV containing one column called "Sequence" or "pep_seq" containing peptide sequences, and one column called "Modifications" or "pep_var_mod" containing their corresponding variable modifications (if detected) in either Mascot or MaxQuant format. This input file may be either an unaltered Mascot output csv, or MaxQuant evidence.txt file, or a user-created file containing only the relevant columns. The Demodifier first parses the input data, extracting the number of modifications for each peptide (Fig 3 step A). It then calls the Unipept API [40] using the pept2lca function to gather the LCAs for each input peptide (Fig 3 step B), before generating all possible deamidation-induced permutations of that input sequence for glutamine (Q), asparagine (N), glutamic acid (E) and aspartic acid (D)

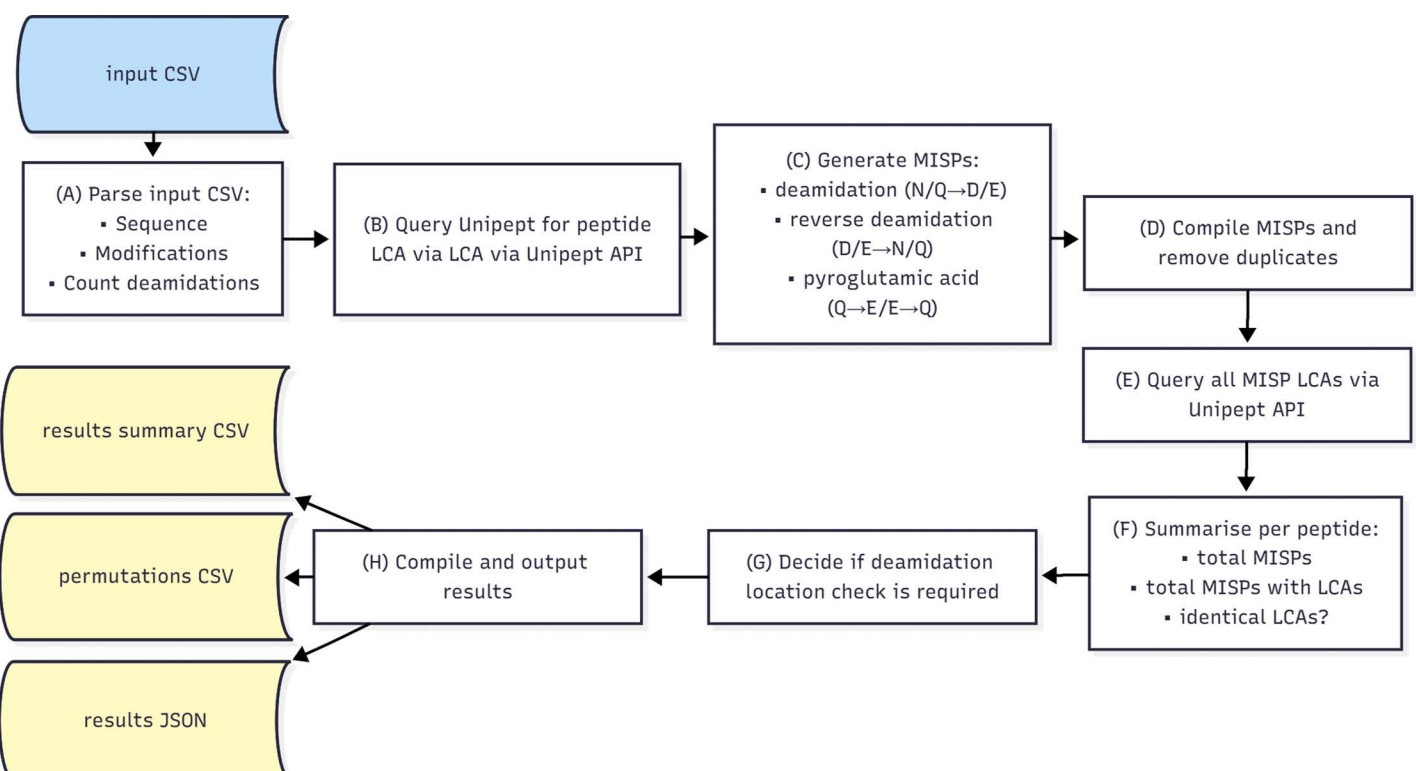

**Fig 3. Schematic overview illustrating The Demodifier's pipeline.** Figure created in Mermaid.live.

residues, and all possible pyroGlu induced permutations for N-terminal glutamine (Q) and glutamic acid (E) (Fig 3 step C, see also further details below). The Demodifier then removes any duplicate MISPs (Fig 3 step D) before querying the Unipept API to gather the LCAs of each MISP (Fig 3 step A). The Demodifier summarises the total number of MISPs, total number of MISPs which yield LCAs and whether the input peptides resultant MISPs yield identical LCAs (Fig 3 step F) and calculates which input peptides require manual deamidation location checking to assess which LCAs are supported (Fig 3 step G). Finally, the results are output in two CSV files and a JSON described below (Fig 3 step H). The researcher is then able to quickly detect input peptides with additional LCAs by consulting the results CSV, before returning to standard palaeoproteomic analysis process (see below). The script incorporates retry logic for transient failures and manages API sessions through persistent, per-thread handling. It is multi-threaded, executing tasks concurrently for optimised performance, and allows the user to specify the number of processor threads. Please note, version 1.4.0 was used in this analysis. It is functionally the same as the most updated version (1.6.0), which was improved to enable additional input file types, and refactored for user clarity following reviewer suggestions.

**Output.** The Demodifier produces a results summary CSV, a permutations CSV and a results JSON. The results summary CSV contains all input peptides and their modifications, annotated with their input LCA, their total number of MISPs, the number of MISPs that yielded LCAs, the specific MISPs that yielded LCA results, their resulting LCAs, whether all resulting LCAs are identical and whether inspection of peptide fragment coverage is required to determine which permutations are supported by a particular peptide spectral match (PSM). The permutations CSV contains the input peptide, modifications, each MISP generated, and each MISP's LCA where they were available. The JSON contains all the above information in a nested structure.

**Permutation generation.** The Demodifier simulates six potential substitutions (Fig 4). First, The Demodifier simulates all permutations of deamidation of glutamine (Q) and asparagine (N) by replacing these residues with glutamic acid (E) and aspartic acid (D), respectively, up to the number of deamidations detected for a given input peptide by search software, and ignoring N-terminal glutamine (Q) if the pyroGlu modification is present. Only permutations with an identical mass to the input peptide are created: i.e., those with the same number or fewer deamidated residues than the input peptide, given that only deamidated glutamine (Q) and asparagine (N) residues have identical masses to glutamic acid (E) and aspartic acid (D), respectively. For each resultant deamidation permutation, The Demodifier then simulates all possible *reverse* deamidation permutations, excluding positions which were previously altered in the first stage, and N-terminal glutamic acid (E) if the pyroGlu modification was detected. That is, it *reverses* deamidation of glutamic acid (E) and aspartic acid (D), *returning* them to glutamine (Q) and asparagine (N) respectively with no maximum number of substitutions (given that glutamic acid and aspartic acid residues have identical masses to deamidated glutamine and asparagine residues respectively). Finally, for each resultant permutation, The Demodifier then simulates glutamine (Q) to glutamic acid (E) substitutions for peptides bearing N-terminal Q with the Gln→pyroGlu modification detection, and E to Q substitutions for peptides bearing N-terminal E with Glu→pyroGlu modification detection. Any duplicate peptide permutations for a given input peptide are removed prior to downstream analysis.

**Flagging potentially unsupportable LCAs.** Certain MISPS would only be possible if deamidation was detected at a particular position in the sequence during the proteomic search. The Demodifier only considers the peptide sequence when generating MISPs, as opposed to the ion coverage of a particular PSM, and therefore does not know the location of deamidations. As a result, where deamidation was detected by search software at a particular position The Demodifier may generate MISPs at other positions which are not actually possible for a particular PSM. This possibility can only occur when the number of deamidations detected by the search software is less than the number of potentially deamidatable residues (NQ), meaning the deamidations could be in several different locations, some of which may not be supported by a PSMs ion coverage. Only peptides which satisfy this criterion *and* generate multiple MISPs which match real protein sequences could be impacted. To flag these cases, the Demodifer outputs a column in the results CSV flagging the input peptides which require manual verification of possible sites of deamidation (i.e., through scrutiny of MS2 spectra, or

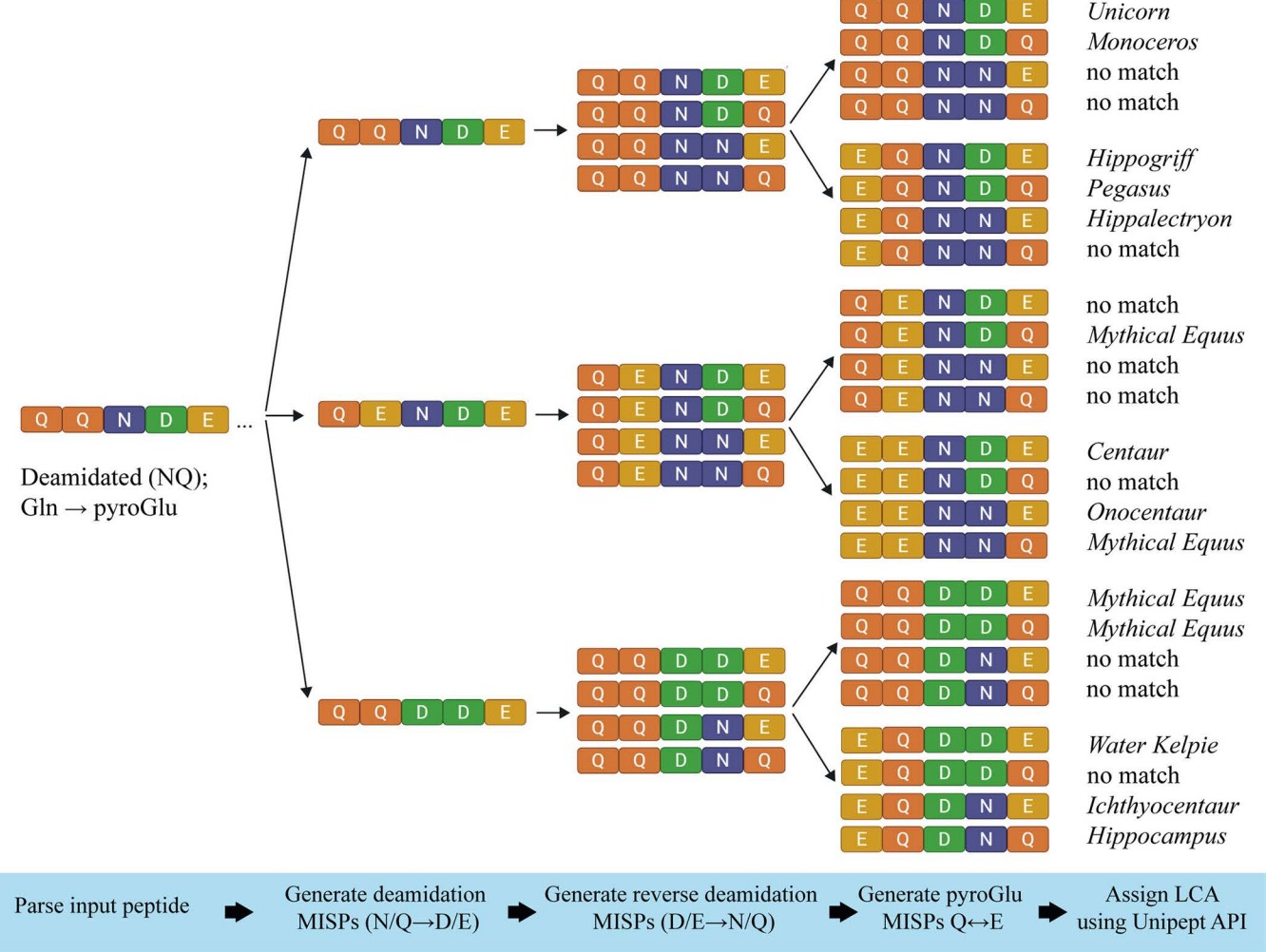

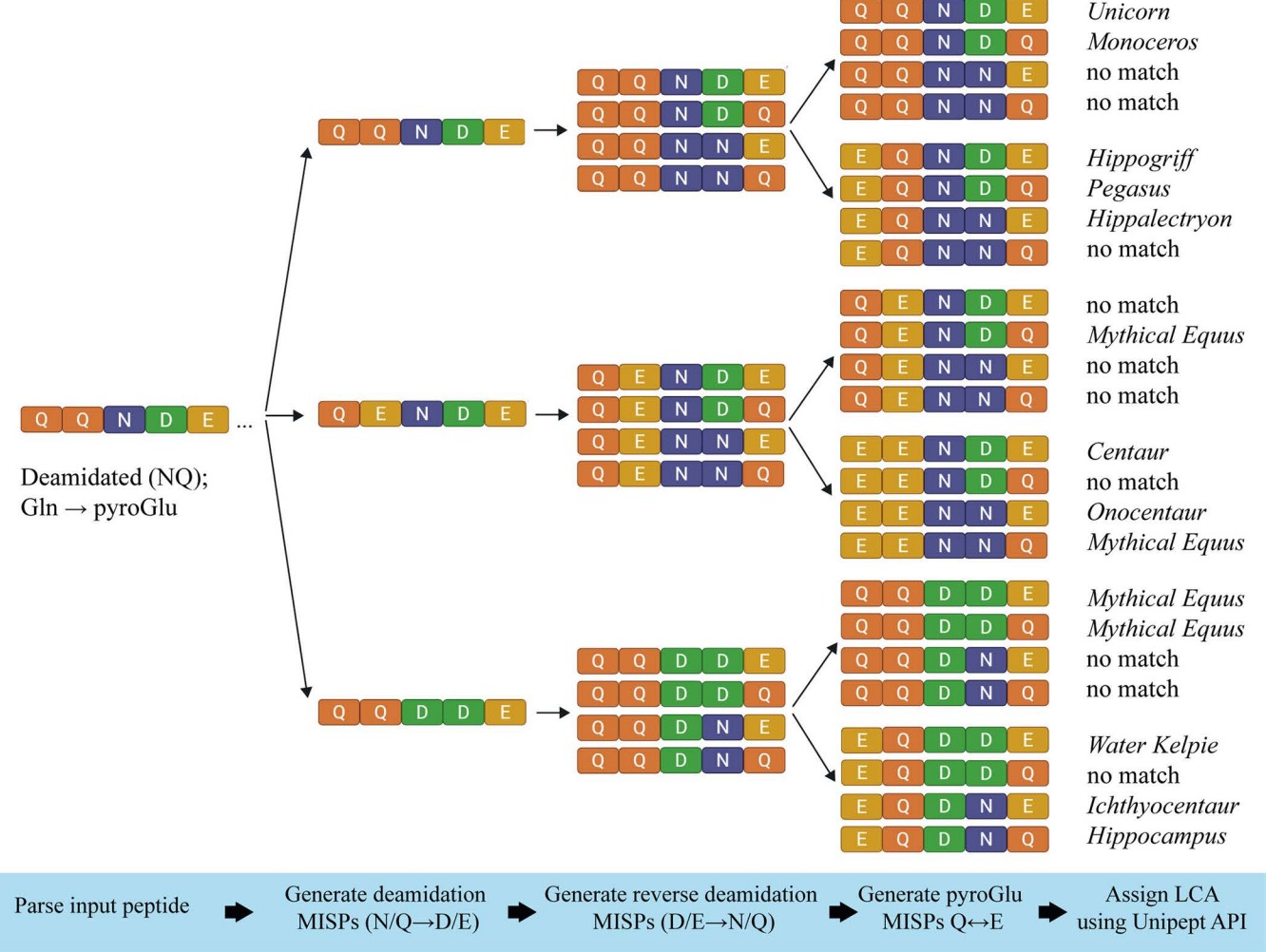

**Fig 4. The Demodifier's process of making sequence permutations for an imaginary peptide.** First, all possible deamidations of N/Q to D/E are simulated up to the number detected by search software and ignoring N-terminal Q if the pyroGlu modification was detected, next reverse deamidation permutations of D/E to N/Q are simulated, ignoring previously altered positions and N-terminal E if the pyroGlu modification was detected, and then if N-term pyroGlu formation was detected, Q to E or E to Q substitutions are simulated depending on starting amino acid. Peptide permutation duplicates are removed (if present) before LCA is assigned for each MISP using Unipept pept2LCA. In the case of this imaginary peptide, the LCA of the input peptide is Unicorn, however the MISPs yield LCAs matching to several other Mythical Equus. Figure created in Biorender.com.

consulting deamidation locations and probabilities if these are provided by the search software). Peptides are flagged as requiring further investigation if the number of deamidations in the "Modifications" column is less than the total number of asparagine (N) and glutamine (Q) residues in a given input peptide (excluding N-term Q if a pyroGlu modification was detected), and the number of deamidations detected is not zero, and the resulting MISPS generate multiple different LCAs which are not identical.

**Usage within the palaeoproteomic workflow.** In a usual palaeoproteomic workflow, the raw mass spectrometry data are first searched using database or *de novo* methods to reveal matching peptide sequences. These are then filtered to exclude laboratory and instrument contaminants before the taxonomic LCAs of the remaining peptide matches are verified using tools such as NCBI protein-protein BLAST. The Demodifier should be used after the initial database or *de*

*novo* search and contaminant filtering steps. Where The Demodifier flags peptides which require checking of deamidation locations to exclude unsupported MISPs, this should then be done. Following this, the researcher should return to the standard workflow, scrutinising the LCA of each peptide (and newfound MISP) following standard practice using NCBI protein-protein BLAST, before interpretation of overall protein LCA and archaeological interpretation.

### Test dataset

A test dataset consisting of peptides and modifications previously detected in archaeological samples was assembled to explore The Demodifier's ability to detect modification-induced sequence variations and their resulting taxonomies. The test dataset consists of all unique peptides reported in ancient protein studies of dental calculus and vessels which are readily accessible (see S1 File), and which include corresponding peptide modifications on a per-peptide basis. The dataset was filtered to exclude identical peptide and modification combinations (with PTMs other than deamidation and pyroglutamic acid formation excluded), and sequence and modification terminology was converted to Mascot format (S2 File). The resulting dataset consisted of 1076 unique peptide and modification combinations drawn from 19 published studies [1–3,23,24,26,28,41–52] and my PhD thesis [39]. The PRIDE accession numbers for these datasets can be found in S3 File (where they were available). Of these 20 datasets, 19 included deamidation (NQ) as a variable modification in their search parameters, while nine included pyroglutamic acid formation from glutamine (Q), and three included pyroglutamic acid formation from glutamic acid (E), (S3 File).

The focus of the case study was to assess the utility of The Demodifier at detecting all deamidation and pyroglutamic acid-based MISPs and their LCAs from a large dataset of archaeologically plausible peptides, rather than to re-assess the taxonomic interpretations of particular PSMs from specific studies. Therefore, the original peptide ion coverages were not consulted, and no MISPs were excluded from this discussion as all would be potentially possible, given different PSMs of the same peptide.

I made the decision not to directly compare The Demodifier's results to the authors' attributed peptide and protein LCAs because this would be an uneven comparison for several reasons. Firstly, most authors assess taxonomy using NCBI protein-protein BLAST while The Demodifier uses Unipept pept2LCA. By virtue of their different algorithms and databases, these can yield differing LCA results. Secondly, as the underlying databases used by these tools change over time, so too do the LCAs they generate. Lastly, researchers use varying logic to assess and present LCAs, for example, sometimes excluding taxa which are not present at their particular site. For these reasons, the most equal and transparent way to assess the utility of The Demodifier at detecting MISPs is to compare the LCA of each peptide before and after The Demodifier has been applied.

## Results and discussion

### How common were modification induced sequence permutations?

Modification-induced sequence permutations were identified for 80.9% of input peptides, with 870 of the 1076 input peptides generating at least one MISP, meaning the remaining 206 input peptides contained no D/E, deamidated N/Q or pyroGlu residues (Fig 3). A total of 14629 MISPs were generated from the 1076 input peptides (S4 and S5 Files). The median number of MISPs generated for an input peptide was 4, with a mode of 2 and mean of 13.5, although the mean was substantially increased by one input β-casein peptide, FQSEEQQQTEDELQDKIHPFAQTQ + 3 deamidated (NQ), which yielded 4096 MISPs (only one of which matched to a real protein sequence).

### How frequently did modification induced sequence permutations result in multiple taxonomic matches?

The results revealed that 16.6% of peptides (179 of 1076) generated at least one MISP which resulted in a taxonomic identification different from that of the input peptide (Fig 5A). Only 18 of these have been reported previously (1.7%)

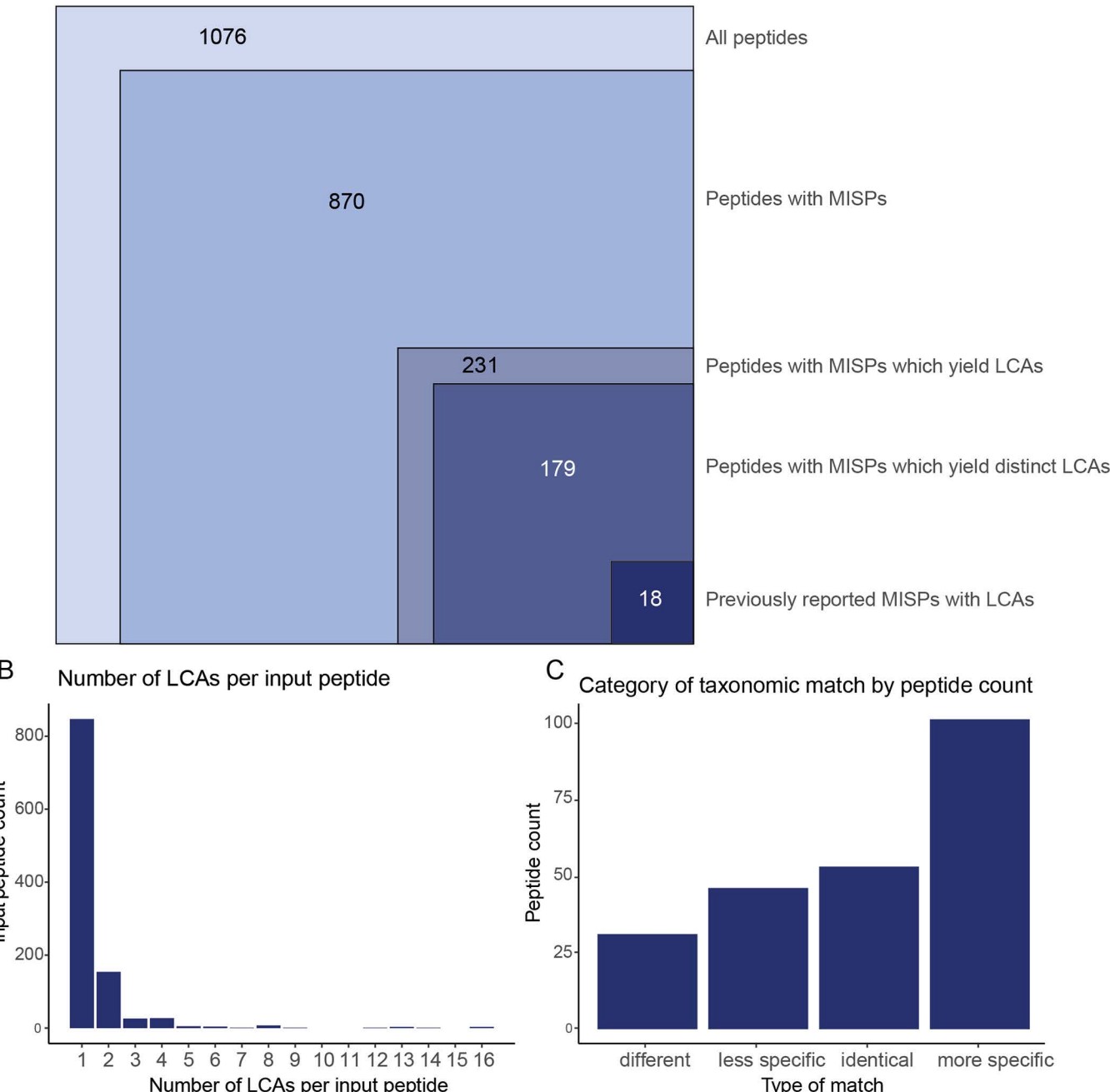

**Fig 5. Graphical overview of The Demodifier's output for the test dataset. (A)** Proportional figure showing the total number of peptides in the test dataset, number of peptides with MISPs, number of peptides with MISPs that match to real protein sequences, those which additionally match to multiple distinct sequences, and those which have previously been reported, **(B)** Number of LCAs per input peptide by input peptide count **(C)** MISP LCA categories by input peptide count including: LCAs which were different to the input peptide's LCA, identical to the input peptide's LCA, less taxonomically specific than the input peptide's LCA, or more taxonomically specific than the input peptide's LCA.

meaning that modification induced alternate peptide taxonomy occurred almost ten times more frequently than previously reported- an increase of nearly 15 percentage points. While some peptides had multiple MISPs yielding distinct taxonomies in addition to their input peptide taxonomy, most only had one additional taxonomic match (Fig 5B). The vast majority of MISPs yielding LCAs resulted from deamidation (220 of 231), while a small minority was caused by a pyroglutamic acid modification (11 of 231). This is probably largely due to the fact that deamidation can occur multiple times in a peptide, while pyroglutamic acid is only formed at the N-terminus. However, it could also be because fewer studies in the test dataset included the formation of N-terminal pyroglutamic acid as a variable modification in their search parameters (9 of 20), compared to deamidation (19 of 20) (S3 File).

### What was The Demodifier's run time?

The Demodifier's mean run time on the test dataset using 8 processors was 7.003 seconds (range = 6.91 to 7.15 seconds, number of replicates = 10). A minimal test was undertaken to assess whether run time would increase in a linear or exponential fashion with increased input size (S6 File). This revealed that run time increased approximately linearly with increased peptide number input, meaning that The Demodifier would be applicable to larger datasets than that tested here. I note, however, that the Unipept API response time varies which can impact total run time.

### Archaeological implications of modification-induced alternative peptide taxonomy

When scrutinised, it was apparent that some, but not all, of the 231 input peptides with MISPs which yielded LCAs could lead to inaccurate archaeological interpretations. Excluding the 52 MISPs which rendered identical LCAs to the input sequence (S7 File), the remaining 179 peptides with MISPs could be classified into three categories (Fig 5C, S7 File):

1) Peptides that could equally be derived from two or more totally different taxonomies

2) Peptides that could be more taxonomically specific than originally thought

3) Peptides that could be less taxonomically specific than originally thought

Of the input peptides with multiple LCA matches, if they had at least one "less specific" match, they were categorised into the third category, even if they also had "more specific" matches, given the more serious implications of less specific taxonomy on an archaeological interpretation. A multiclass confusion matrix comparing input peptide LCA to MISP peptide LCAs can be seen in S8 File.

**Peptides yielding different LCAs to input peptide.** The first category included peptides which could belong to two completely different taxonomic lineages, as is the case for the well-known peptide TPEVD**D/N**EALEK, which could equally be derived from *Ovis* or Bovinae βLG. This class consisted of 31 peptides derived from 14 proteins (S7 File). Sometimes, such peptides would be easily detectable because they initially appear to belong to an archaeologically unlikely taxonomy, such as the haemoglobin peptide VDQVGAEALGR (deamidated) detected by Evans [39], which initially matched to *Triturus cristatus* (Northern crested newt), but has a MISP which matches to Laurasiatheria (a superorder of placental mammals which includes all true insectivores, bats, carnivorans, pangolins, even-toed ungulates and odd-toed ungulates). Conversely, sometimes this category of MISPs produce additional LCAs which would be unlikely in most archaeological contexts. For example, the peptide βLG TKIPAVFKID, which initially matches to Bovidae, but also has a MISP matching specifically to *Physeter catodon* (Sperm whale). However, other examples are archaeologically plausible and could easily be missed because their input peptide LCAs seem perfectly reasonable. For example, the βLG peptide IPAVFKIDALNEN + 2 Deamidated (NQ) (Also detected by Evans [39]), initially appears to match Pecora (an infra-order of even-toed hoofed ruminants), but also has a MISP which matches to *Sus scrofa*. This class of peptides can result in potentially spurious taxonomic detections, which could impact archaeological interpretation. Ancient samples often yield only a few peptides of a particular protein, of which fewer still are likely to be taxonomically specific. In such

cases, unscreened MISPs could alter archaeological interpretations by leading the researcher to believe that a potentially spurious taxon was utilised on their site, causing them to miss evidence for a taxon which was actually present.

**Peptides yielding more specific LCAs than input peptide.** The second class of MISPs includes peptides that could potentially be more taxonomically specific than indicated by the input peptide LCA (S7 File). This class comprised 101 peptides derived from 60 proteins. For instance, the α-lactalbumin peptide NNGKTEYGLFQINNK + 3 Deamidated (NQ) initially appears taxonomically conserved, matching to Laurasiatheria, however The Demodifier revealed a MISP belonging specifically to *Equus caballus* (Horse), a species for which the archaeological sample bore other peptide matches [42]. It's worth noting that given the higher number of detected deamidations than deamidatable residues in this peptide, scrutiny of deamidation locations would be required to confirm whether this MISP LCA was supported in each specific PSM. Another example is the βLG region VYVEELKPTPE (identified in seven of the test dataset studies), which initially has an LCA of Pecora, but in fact generates several MISPs matching to more specific taxonomies (*bos taurus, Moschus moschiferus* and *Odocoileus virginianus texanus).* Peptides from plant proteins were also identified in this category, including the 12S seed storage globulin 1 peptide ALPVDVLANAYR (unmodified) detected by Hendy et al. [26], which initially matches to Magnoliopsida (a large class of flowering plants) but has a MISP matching specifically to *Avena sativa* (Oat) (although at the time of their study, both MISPs would have matched to *Avena* due to subsequent database additions). Another example is the ɣ-hordein-1 peptide VMQQQCCLQLAQIPEQYK + 2 Deamidated (NQ), detected by Hendy et al. [3], which initially matches to Triticeae (a tribe of grasses which includes many domesticated grains including wheat, barley and rye) but has a MISP matching specifically to *Hordeum vulgare subsp. vulgare* (domesticated barley); a taxon for which there was additional proteomic evidence in this sample. This is another example of a MISP LCA which would require scrutiny of deamidation locations to confirm which LCA or LCAs were supported by a specific PSM.

This class of MISPs may not change archaeological interpretation as the researcher can't exclude the possibility that the peptide only bears evidence for the less specific taxonomy. However such findings may offer additional insight when viewed in the context of other peptides of a given protein in an ancient sample. For example, the α-S1 casein peptide region IGVNQELAYFYP**E**LFR (unmodified), detected by Tanasi et al. [52] and Siano et al. [49] in Maltese and Italian contexts respectively, initially appears to belong to Bovinae but has a MISP (IGVNQELAYFYP**Q**LFR) which belongs specifically to *Bubalus Bubalis* (water buffalo). Similarly, the frequently detected βLG peptide VLVLDTDYKK (unmodified), which was reported in 13 of the test-dataset studies, initially appears to belong to Pecora, yet also has a MISP belonging to *Bubalus Bubalis*. Water buffalo are a major dairy animal, meaning this finding would be important in archaeological contexts where they are present, especially considering that water buffalo βLG only has one species-specific tryptic peptide, and it is not in a sequence region that has frequently been detected.

**Peptides yielding less specific LCAs than input peptide.** Conversely, the third class of MISP taxonomies are those that are less taxonomically specific than the input peptide taxonomy. In the test dataset, this class consisted of 46 peptides derived from 30 different proteins (S7 File). These examples have the potential to change archaeological interpretations by reducing the taxonomic specificity of detected proteins, meaning that archaeological evidence for more specific (but potentially inaccurate) taxa may be removed. However, I note that in the test dataset, the samples often had additional taxonomically-specific peptides, meaning the protein taxonomy would not have changed. For example, while it initially appears that the α-amylase inhibitor peptide SGPWSWCNPATGYK + Deamidated (NQ) detected by Tanasi et al. [52] matches *Triticum aestivum* (Common wheat), The Demodifier revealed a MISP with an LCA of Triticeae (a tribe of grasses which includes many domesticated grains such as wheat, barley and rye). Similarly, the β-casein peptide GVAGEPGRNGLPGGPGLR + Deamidated (NQ) detected by Siano et al. [49] initially appears to match *Bos taurus* (cattle) but in fact has a MISP with the LCA of Artiodactyla (an order of even-toed ungulates which contains most domestic and several wild species). Another example is the 11S globulin isoform 4 peptide QEEEPYYGR (unmodified) detected by Scott et al. [1], which initially matches *Sesamum indicum* (Sesame) but has a MISP belonging to Eukaryota. While, as mentioned above, in each of these studies the presence of additional species-specific peptides in the sample meant that

the protein taxonomic assessment would not have been altered, if such peptides had been the lynchpin in a protein's taxonomic assessment, this could have impacted the archaeological interpretation.

As with the first category of matches, occasionally matches of this class may be readily detectable due to archaeologically unlikely taxonomic matches yielded by the input peptides. For instance, Evans [39] detected the Haemoglobin subunit beta peptide QEVGGEALGR + Gln→pyroGlu in a Romano-British vessel that initially appears to belong to *Hippopotamus amphibius* (Hippopotamus), which is highly improbable in this context. Upon further inspection this peptide generated a MISP matching to Sarcopterygii (a clade of vertebrates which contains all tetrapods). As can be seen, this class of MISPs could have serious implications on archaeological interpretation of proteomic results, as they may reveal the initial peptide taxonomy to be inaccurately specific.

This analysis revealed that The Demodifier was effective at screening for MISPS in the test dataset, which were shown to be nearly ten times more common than previously reported. The results revealed that modification-induced alternate peptide taxonomies could potentially be pivotal in archaeological analyses, as they can completely change taxonomic detections or provide more (or less) specific taxonomic detections than initially thought. I reiterate that the risk of incorrect taxonomic matches due to MISPs would have been alleviated in many cases in the test dataset due to the presence of other taxonomically specific peptides contributing to overall protein LCA in the ancient samples in question. However, this will not always be the case, and since taxonomic specificity is one of the significant advantages of metaproteomics over other analytic methods, recognising these alternate peptide taxonomies in archaeological interpretations is crucial in avoiding spurious or overly specific taxonomic detections.

### Challenges and future directions

**LCA attribution.**  The biggest challenge to The Demodifier is that automated LCA assignment using Unipept pept2LCA can only sometimes capture the nuances of manual BLAST searching. As such, I stress that The Demodifier was designed to screen for MISPs that yielded alternate taxonomies but not to replace manual BLAST-based LCA analysis. When a peptide matches to select (but not closely related) taxonomies, the LCA can appear needlessly broad, particularly where peptide matches to archaeologically impossible or unlikely species are present. Therefore, once peptides bearing MISPs have been detected by The Demodifier, users should further assess taxonomic matches using tools that enable scrutiny of each sequence match (e.g., NCBI protein-protein BLAST GUI). By scrutinising matched protein sequences manually, the user is able to investigate the quality of sequences and their relevance to the context. For example, several bacterial sequences in NCBI NR appear to have become contaminated with common food proteins such as grain protein sequences [1]; or those used in cell culture including milk proteins [43] and ovalbumin [41]. This means that automatic LCAs may be assigned at a much higher taxonomy than is actually accurate. Similarly, sometimes manually scrutinising the BLAST matched sequences allows the user to flag potentially low quality matches. For example, Wilkin et al. [45] note that certain Bovinae and Caprinae blood peptides additionally match to white tailed deer and elk, but only in predicted protein sequences. Lastly, when addressing certain archaeological questions, the taxonomy to which a sequence does *not* match can sometimes be just as meaningful as the taxonomy to which a sequence does match. For example, in their analysis of dietary proteins present in human dental calculus from British Neolithic individuals, Charlton et al. [28] discovered that one sample harboured evidence for βLG derived from *Capra sp.*, as well as evidence for Bovidae excluding *Capra*, meaning that the individual had consumed dairy from multiple distinct taxa. As a result of these challenges, while automating LCA in The Demodifier was necessary to detect permutations with real sequence matches, manually BLAST searching and interrogating the taxonomy trees in the web version of NCBI protein-protein BLAST can yield valuable information, and despite its time-consuming nature should be seen as the gold standard of LCA attribution in ancient protein studies.

The decision to use Unipept rather than a BLAST-based tool was made due both to recent advances in Unipept, and difficulties and limitations of other tools. Previously Unipept was only capable of generating LCA for tryptic peptides

(or those with missed cleavages), meaning that it could not assign LCA for many of the degraded peptides identified in palaeoproteomic studies. However, a recent update allows semi-tryptic and non-tryptic peptides to be matched [53]. While many popular BLAST-based LCA-assignment tools were investigated for use in The Demodifier, none fulfilled the criteria of generating LCA for peptides (rather than whole proteins), on the command line (rather than through a GUI), and which do not require the end user to download the entire NCBI NR/ TrEMBL database or use paid cloud servers (neither of which are accessible to many users).This meant that it was not possible to integrate these BLAST-based tools. Nevertheless, throughout this study it became apparent that the peptide LCA generated using Unipept does not always match the LCA generated by NCBI BLAST searching. This is partly a function of the different databases the two tools use (NCBI protein-protein BLAST uses the NR database by default, while Unipept uses UniprotKB (Swissprot + TrEMBL) by default. As such, cross checking is advisable, and a larger conversation about taxonomic assignment in palaeoproteomics is warranted. I also reiterate that the taxonomic results of the case study are not intended to be compared directly with author-attributed LCAs from the original studies. This would be an inaccurate comparison as, with the addition of new protein sequences to the databases over time and the different tools used for taxonomic assessment, LCAs may have changed since studies were published. Even between the initial submission of this manuscript and its revision, minor changes were observed in the LCAs yielded by the test dataset for a few peptides.

**Additional modifications.** The Demodifier currently screens for deamidation and pyroglutamic acid-formation induced sequence permutations, however it is probable that other PTMs could cause similar issues. For example, Engels et al. [54] very recently created a tool for collagen species classification, which screens for isobaric alanine-to-serine substitutions in addition to deamidation. Future research could expand The Demodifier to detect this and other additional PTMs.

**Probability of modification.** The current version of The Demodifier treats all modification sites as equally probable, despite the fact that certain modifications and modification sites may be more likely than others. For instance, deamidation of glutamine (Q) is less common given that it occurs over a much slower timescale than deamidation of asparagine (N) [8], and residues in flexible regions have been found to be more likely sites of asparagine deamidation [18]. In the case of pyroglutamic acid formation, deamidation of N-terminal glutamine (Q) occurs at a higher rate than dehydration of N-terminal glutamic acid (E), however factors such as temperature, pH, and enzymatic activity also play a role [37,38]. Therefore, future iterations could introduce some form of probabilistic scoring. However, while some modification scenarios may be more likely than others, given the unknown taphonomic and cultural history of many archaeological samples, I decided to employ a conservative approach, allowing The Demodifier to output all MISPs regardless of probability, with the rationale that if there is any possibility that a modification could occur, it should be screened for.

**Unsupported MISPs.** The Demodifier calculates all possible MISPs for a given peptide, regardless of deamidation locations. Therefore, as noted in the methods section, certain MISPs generated by The Demodifier may not be possible for a particular PSM detection and are flagged as requiring manual scrutiny of deamidation locations. In this case study, this issue only impacted 26 of the 1076 unique input peptides (2.4%). One possible solution may be to adapt The Demodifier to accept input peptides with deamidation locations embedded within the sequence and restrict The Demodifier from generating permutations for any other locations, however this is not without its problems. Firstly, depending on which search software is used, in some cases multiple deamidation positions remain possible due to incomplete ion coverage, yet only the most probable is annotated in the peptide sequence in search software outputs. In such cases, excluding all non-annotated positions may exclude MISPs that are actually possible. Secondly, not all software outputs a peptide sequence that is annotated with the modification positions, meaning that this solution may reduce accessibility of the Demodifier to some users. Another approach may have been for The Demodifier to accept fragment mass information and isolate the possible deamidation positions from this, however this would be drastically more complicated and resource-intensive than the current approach. Moreover, it would complicate user experience as, rather than simply inputting a (potentially filtered or curated) list of peptides of interest and their modifications, the user

would have to input all fragment mass data, which may be vast and contain many contaminants. For these reasons and considering the low number of cases requiring scrutiny in the test dataset, I chose not to implement this approach.

**A note on archaeological usage.** An important disclaimer is that The Demodifier is only designed to aid in detection of MISPs and any additional LCAs which they yield. It is not designed to replace any other part of the palaeoproteomic or analysis of interpretation pipeline, including BLAST searching or overall protein LCA interpretation. In particular, while the focus of this analysis was scrutinising MISPs which yielded alternate taxonomies on a per-peptide basis, it is critical to reiterate that in a usual archaeological use-case researchers must then interpret the resulting LCAs of all peptides of a detected protein together, to understand which protein LCAs are supported in their sample.

This study provided an opportunity to explore the utility of The Demodifier across a large dataset of archaeologically detected peptides, which will hopefully garner awareness of the field-wide risk of modification induced alternate peptide LCAs, and equip researchers with a rapid open source screening tool to combat this issue.

## Conclusion

In this paper I present The Demodifier 1.6.0: an open source tool to screen for PTM-induced alternate peptide taxonomies. The results of this study reveal that modification-induced alternate peptide taxonomies are much more common than previously recognised in ancient protein studies. Often these multimorphic peptide sequences are only discovered when unexpected taxonomic hits are unearthed, while "normal" seeming taxonomic results do not always garner scrutiny. Screening of all sequences which could have MISPs is therefore necessary, regardless of initial input peptide taxonomy. The results revealed that modification-induced alternate peptide taxonomy is almost ten times more common than previously reported in ancient protein studies, and can result in peptide permutations with completely different, more specified or less specific taxonomic LCAs compared to the input peptide. It is likely that future proteomic analyses will reveal additional peptides with alternate taxonomies, meaning screening for MISPs should be standard practice in protein studies. The Demodifier provides a simple and efficient open source tool for detecting such sequences, thus avoiding spurious or overly specific taxonomic detections which could yield inaccurate archaeological interpretations.

## Supporting information

**S1 File. All peptides contributing to the test dataset.**
(XLSX)

**S2 File. Input file consisting of all unique peptide and modification combinations.**
(XLSX)

**S3 File. Summary of studies contributing to test dataset.**
(XLSX)

**S4 File. Demodifier results output.**
(XLSX)

**S5 File. Demodifier permutations output.**
(XLSX)

**S6 File. Investigation of impact of input dataset size on Demodifier run time.**
(XLSX)

**S7 File. Table categorising peptide LCAs including those with more specific, less specific and completely distinct taxonomy after use of The Demodifier.**
(XLSX)

**S8 File. Confusion matrix of peptide LCA before and after use of The Demodifier.**
(XLSX)

## Acknowledgments

I would like to thank Matthew Collins for providing valuable feedback on a version of this manuscript, and Oliver Boyd for his assistance in converting the script to an executable. Thank you also to Jessica Hendy and Christiana Scheib for encouraging my work on this paper. I would also like to thank Laura Cristina Viñas Caron, Louise Le Meillour, Meaghan Mackie, Shevan Wilkin and two anonymous reviewers for their extremely helpful reviews on the PCI platform, as well as the two helpful anonymous reviewers for PLOS ONE.

## Author contributions

**Conceptualization:** Miranda Evans.

**Data curation:** Miranda Evans.

**Formal analysis:** Miranda Evans.

**Funding acquisition:** Miranda Evans.

**Investigation:** Miranda Evans.

**Methodology:** Miranda Evans.

**Project administration:** Miranda Evans.

**Resources:** Miranda Evans.

**Software:** Miranda Evans.

**Validation:** Miranda Evans.

**Visualization:** Miranda Evans.

**Writing – original draft:** Miranda Evans.

**Writing – review & editing:** Miranda Evans.

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
