## [Decision Letter · Decision Letter 0]

2 Oct 2025

Dear Dr. Evans,

Thank you for submitting your manuscript to PLOS ONE. After careful consideration, we feel that it has merit but does not fully meet PLOS ONE’s publication criteria as it currently stands. Therefore, we invite you to submit a revised version of the manuscript that addresses the points raised during the review process.

We look forward to receiving your revised manuscript.

Kind regards,

Enrico Greco

Academic Editor

PLOS ONE

Journal Requirements:

2. Please note that PLOS One has specific guidelines on code sharing for submissions in which author-generated code underpins the findings in the manuscript. In these cases, we expect all author-generated code to be made available without restrictions upon publication of the work. 

Please review our guidelines at https://journals.plos.org/plosone/s/materials-and-software-sharing#loc-sharing-code and ensure that your code is shared in a way that follows best practice and facilitates reproducibility and reuse.

3. We noted in your submission details that a portion of your manuscript may have been presented or published elsewhere:

“This article was preprinted and recommended by PCI. This does not constitute dual publication as PCI is a transparent review platform, not a journal.

All reviews, responses, and recommendation can be found here: https://archaeo.peercommunityin.org/PCIArchaeology/articles/rec?id=600”

4. Please note that your Data Availability Statement is currently missing the repositories name. If your manuscript is accepted for publication, you will be asked to provide these details on a very short timeline. We therefore suggest that you provide this information now, though we will not hold up the peer review process if you are unable.

Reviewers' comments:

Reviewer's Responses to Questions

**Comments to the Author**

1. Is the manuscript technically sound, and do the data support the conclusions?

Reviewer #1: Yes

Reviewer #2: Yes

2. Has the statistical analysis been performed appropriately and rigorously?

Reviewer #1: N/A

Reviewer #2: N/A

3. Have the authors made all data underlying the findings in their manuscript fully available?

Reviewer #1: Yes

Reviewer #2: Yes

4. Is the manuscript presented in an intelligible fashion and written in standard English?

Reviewer #1: Yes

Reviewer #2: Yes

Reviewer #1: The author reports the implementation of a useful tool, called The Demodifier, designed to help in accurate archaeological interpretation of proteomics data. The tool is designed to screen for possible alternatives to the sequence assigned by the MS/MS search engine, focusing on two main modifications: deamidation of asparagine and glutamine, which may be confused for unmodified aspartic and glutamic acid, and cyclization at the N-terminus, which renders glutamine indistinguishable from glutamic acid. The Demodifier does not analyze MS/MS spectra directly, but screens for modification-induced sequence permutations detecting all peptide variants which may originate from a list of identified sequences. The tool has been tested on several publicly available data sets. After some minor concerns are addressed, I would support the publication of the manuscript.

L284: the argument is not convincing. At least for some of the studies under consideration, would it be possible to assess the utility of the The Demodifier in supporting LCA assignment ?

L302: Most important point: the manuscript states “The results revealed that 16.6% of peptides (179 of 1076) generated at least one MISP which resulted in a taxonomic identification different from that of the input peptide. Only 18 of these have been reported previously, meaning that modification induced alternate peptide taxonomy is 15% more common than currently reported.” I believe that the number is obtained by the difference between 179/1076=16.6% and 18/1076 = 1.6%. In this case, the author should state that alternate peptide taxonomy is "15 percentage points higher than…."which is different from "15 percent higher". The statement is present more than once in the manuscript and in the abstract as well.

Usefulness: The author could state more clearly how this tool should be integrated in a workflow of archeoproteomic data analysis.

Reviewer #2: The present article, titled "The Demodifier: a tool for screening modification-induced alternate peptide taxonomy in palaeoproteomics" is innovative and will be beneficial for the paleoproteomics community. This tool helps with data validation and can solve some taxonomy misclassifications issues.

This article can be published in PLOS One with minor revisions:

Introduction:

- line 49: Why does the author consider "a mass shift of approximately +0.984 Da"? In the case of approximateely value, what is the precision (±)? Or is it the exact mass shift?

- line 80: Please write Latin words in italics, such as Bovinae and others, throughout the document

- line 80: Please write Grec symbol in italics, such as β and others, throughout the document

- Fig. 1: Please add "protein" on both sides of amino acids and their derivatives, as in Fig. 2, for a better understanding that it is proteinogenic amino acids.

- Fig. 2: please add the note about isoAsp and isoGlu are not depicted as Fig. 1

The Demodifier:

- line 163: I ran the Demodifier 1.4 executables from your Zenodo repository with Python3, and also imported it and followed your tutorial on your GitHub repository; however, it did not work. Why did the author develop it in Python when the paleoproteomics community, composed mainly of archeologists and chemists, used R? A detailed video tutorial will be beneficial for non-Python experts.

- line 170: Please spell MaxQuant with Q throughout the document

- line 167-170: Does the user have to create itself a .cvs file with "Sequence" and "Modifications" columns? Why the script does not require unmodified Mascot or MaxQuant output files (such as evidence.txt MQ file), and the script extracts Sequence and Modifications information of interest with a few additional commands? Automating data treatment will facilitate the user experience for non-experts.

- Fig. 3, box C: The "Q↔E" in Chemistry means a reversible conversion of Q to E. Do you mean "Q/E → pyroGlu" ?

- line 393: As you used β for some proteins, please use ɣ-hordein-1 instead.

- line 404: Same with ɑ

- line 393: Please add the UniProt code for each protein mentioned.

Probability of modification:

- line 530: Probabilistic scoring is a great idea! I hope it can be developed in the future

- line 529: Please delete the double ")"

Data availability:

- Are you planning to add an MIT licence to your GitHub repository?

Acknowledgment:

- Are they reviewers from the peer-review Community platform?

**Do you want your identity to be public for this peer review?** For information about this choice, including consent withdrawal, please see our Privacy Policy

Reviewer #1: No

Reviewer #2: No

---

## [Author Response · Author response to Decision Letter 1]

4 Dec 2025

Please see my response to reviews below. I have also provided this as a PDF, where I have responded in red, which may make it easier to differentiate my response from the reviewers' suggestions.

Review Response: The Demodifier: a tool for screening modification-induced alternate peptide taxonomy in palaeoproteomics

I would like to thank the reviewers for the thoughtful and considered reviews.

For clarity, I have responded in red (in the PDF version). The line numbers refer to the marked-up version of the manuscript.

Reviews:

-Reviewer #1: The author reports the implementation of a useful tool, called The Demodifier, designed to help in accurate archaeological interpretation of proteomics data. The tool is designed to screen for possible alternatives to the sequence assigned by the MS/MS search engine, focusing on two main modifications: deamidation of asparagine and glutamine, which may be confused for unmodified aspartic and glutamic acid, and cyclization at the N-terminus, which renders glutamine indistinguishable from glutamic acid. The Demodifier does not analyze MS/MS spectra directly, but screens for modification-induced sequence permutations detecting all peptide variants which may originate from a list of identified sequences. The tool has been tested on several publicly available data sets. After some minor concerns are addressed, I would support the publication of the manuscript.

I thank Reviewer 1 for their helpful review; I am delighted that you found The Demodifier to make a valuable contribution to the palaeoproteomic community.

-L284: the argument is not convincing. At least for some of the studies under consideration, would it be possible to assess the utility of the The Demodifier in supporting LCA assignment?

Thank you for this suggestion- it is one that I thought deeply about when planning the research design.

The utility of The Demodifier in supporting LCA assignment is assessed by comparing the input peptide’s original LCA result to The Demodifier’s LCA results for that peptide. This is undertaken on a dataset containing all unique peptides from 20 studies on dietary proteins in calculus and vessel residues.

Unfortunately, your suggestion that I could compare the Demodifier’s LCAs to the author- attributed LCAs of each PSM they detected, would be a dangerous and unequal comparison, as use of the Demodifier is not the only variable which has changed since the author’s made their LCA assessments.

Firstly, due to the expanding nature of the databases commonly used for LCA attribution (Uniprot KB/TrEMBL and NCBI NR), LCAs change over time as more protein sequences are added or removed. Indeed, as stated in the text (line 597), even some of the peptide LCAs generated in this study changed between upload of the preprint, and resubmission to the PCI review platform. Secondly, directly comparing LCAs generated by The Demodifier to author-attributed LCAs would not be possible as the authors of the studies in the test dataset almost all used NCBI protein-protein BLAST to assess peptide LCA while The Demodifier uses Unipept pept2LCA. Given their differing underlying algorithms and databases, these tools not infrequently yield different LCAs. Lastly, when interpreting peptide LCAs, different authors use different logic in how they assess and present LCA results. Some authors provide only the overall Lowest Common Ancestor for the peptide, while others break down matches further, sometimes specifying which matches are archaeologically possible/improbable, still others flag relevant taxa to which a peptide did not match in the LCA results, as this sometimes aids in archaeological interpretations. A one-for-one comparison of The Demodifier-generated LCAs with the original author’s LCAs would therefore be unequal and dangerous, given the many variables at play. Due to these reasons, the fairest and most transparent way to assess the utility of The Demodifier to detecting MISPs is to compare the LCA of each peptide before and after The Demodifier has been applied, as is done in the text.

To clarify these reasons, I have added the following sentence in the text at line 341:

“I made the decision not to directly compare The Demodifier’s results to the authors’ attributed peptide and protein LCAs because this would be an uneven comparison for several reasons. Firstly, most authors assess taxonomy using NCBI protein-protein BLAST while The Demodifier uses Unipept pept2LCA. By virtue of their different algorithms and databases, these can yield differing LCA results. Secondly, as the underlying databases used by these tools change over time, so too do the LCAs they generate. Lastly, researchers use varying logic to assess and present LCAs, for example, sometimes excluding taxa which are not present at their particular site. For these reasons, the most equal and transparent way to assess the utility of The Demodifier at detecting MISPs is to compare the LCA of each peptide before and after The Demodifier has been applied.”

-L302: Most important point: the manuscript states “The results revealed that 16.6% of peptides (179 of 1076) generated at least one MISP which resulted in a taxonomic identification different from that of the input peptide. Only 18 of these have been reported previously, meaning that modification induced alternate peptide taxonomy is 15% more common than currently reported.” I believe that the number is obtained by the difference between 179/1076=16.6% and 18/1076 = 1.6%. In this case, the author should state that alternate peptide taxonomy is "15 percentage points higher than…."which is different from "15 percent higher". The statement is present more than once in the manuscript and in the abstract as well.

Thank you for catching this; you are quite right that the original phrasing “15% higher” was inaccurate, as the value represents an increase of about 15 percentage points (from 1.7% to 16.6%), not a relative 15% increase.

Previously, MISPs yielding additional taxonomic matches had been reported for 18 of 1076 peptides (1.7%), whereas the Demodifier identified such cases for 179 of 1076 peptides (16.6%), which is 9.8 times higher. This corresponds to an absolute increase of nearly 15 percentage points, as you rightly noted.

To ensure accuracy and clarity, I have revised the manuscript text (line 368) to read:

"The results revealed that 16.6% of peptides (179 of 1076) generated at least one MISP which resulted in a taxonomic identification different from that of the input peptide (Fig. 5A). Only 18 of these have been reported previously (1.7 %) meaning that modification induced alternate peptide taxonomy occurred almost ten times more frequently than previously reported- an increase of nearly 15 percentage points"

I have also amended the incorrect value throughout the text at lines 28, 534, and 668.

-Usefulness: The author could state more clearly how this tool should be integrated in a workflow of archeoproteomic data analysis.

Thank you, this is a good point, I have added the following paragraph at line 306:

“Usage within the palaeoproteomic workflow

In a usual palaeoproteomic workflow, the raw mass spectrometry data are first searched using database or de novo methods to reveal matching peptide sequences. These are then filtered to exclude laboratory and instrument contaminants before the taxonomic LCAs of the remaining peptide matches are verified using tools such as NCBI protein-protein BLAST. The Demodifier should be used after the initial database or de novo search and contaminant filtering steps. Where The Demodifier flags peptides which require checking of deamidation locations to exclude unsupported MISPs, this should then be done. Following this, the researcher should return to the standard workflow, scrutinising the LCA of each peptide (and newfound MISP) following standard practice using NCBI protein-protein BLAST, before interpretation of overall protein LCA and archaeological interpretation.”

Given the addition of this paragraph, I have removed the previous, much briefer sentence outlining this information from the paragraph before to avoid repetition.

- Reviewer #2: The present article, titled "The Demodifier: a tool for screening modification-induced alternate peptide taxonomy in palaeoproteomics" is innovative and will be beneficial for the paleoproteomics community. This tool helps with data validation and can solve some taxonomy misclassifications issues.

Thank you Reviewer 2, I am very happy that you find that The Demodifier will benefit the palaeoproteomic community, and I thank you for your helpful review.

-This article can be published in PLOS One with minor revisions:

Introduction:

line 49: Why does the author consider "a mass shift of approximately +0.984 Da"? In the case of approximateely value, what is the precision (±)? Or is it the exact mass shift?

Thank you, this is the exact monoisotopic mass shift to 3 decimal places. I have corrected the text at line 49 in line with your comment to:

“resulting in a monoisotopic mass shift of +0.984 Da (3 d.p.)…”

- line 80: Please write Latin words in italics, such as Bovinae and others, throughout the document

Thank you for your suggestion. The taxonomic nomenclature formatting in the manuscript currently follows the guidelines set out by the International Code of Zoological Nomenclature (ICZN), which state that:

“The scientific names of genus- or species-group taxa should be printed in a type-face (font) different from that used in the text; such names are usually printed in italics, which should not be used for names of higher taxa.” (International Trust for Zoological Nomenclature 2000, Appendix B.6).

I note that the PLOS One guide for authors refers only to italicisation of species and genus (https://journals.plos.org/plosone/s/submission-guidelines), and that recent articles published in PLOS One do not italicise taxa above genus. For example:

“Potato (Solanum tuberosum L.) is a tuber belonging to the Solanaceae family and an important food source in many regions.” (Gao et al. 2025)

“Pterygoplichthys disjunctivus [1] (common name: Vermiculated Sailfin Catfish, Armored Sailfin Catfish, Janitor fish, or Crocodile Catfish) belongs to the Family Loricariidae, Order Siluriformes” (Sharma et al. 2025)

“In this study, the mitochondrial genomes of five species such as Episparis tortuosalis, Pandesma quenavadi, Erebus macrops, Polydesma boarmoides and Xanthodes albago from two families in the superfamily Noctuoidea were sequenced, assembled, and annotated.” (Kuppusamy et al. 2025)

Following this precedent, subfamilies such as Bovinae and other taxonomic levels above genus have not been italicised in this manuscript.

- line 80: Please write Grec symbol in italics, such as β and others, throughout the document

Thank you, I have now made the requested correction throught the text

- Fig. 1: Please add "protein" on both sides of amino acids and their derivatives, as in Fig. 2, for a better understanding that it is proteinogenic amino acids.

Thank you, this is a good point, I have now made the requested correction to Figure 2.

- Fig. 2: please add the note about isoAsp and isoGlu are not depicted as Fig. 1

Thank you, I have added this at line 131

The Demodifier:

- line 163: I ran the Demodifier 1.4 executables from your Zenodo repository with Python3, and also imported it and followed your tutorial on your GitHub repository; however, it did not work. Why did the author develop it in Python when the paleoproteomics community, composed mainly of archeologists and chemists, used R? A detailed video tutorial will be beneficial for non-Python experts.

Thankyou for your comment. My sincerest apologies that the Demodifier did not work for you. I completely agree that a video tutorial would be a great way to increase user accessibility, particularly for those who are not familiar with Python. I will make this a priority for the near future.

To answer your question, The Demodifier was written in Python because it is commonly used by the palaeoproteomic community for creating bioinformatic tools. For example, the palaeoproteomic tools ClassiCOL (Engels et al. 2025), DeamiDATE (Ramsøe et al. 2020) and Proteoparc (Carrillo-Martin et al. 2025) are all written in Python. This reflects broader trends within the modern proteomic community, which is largely Python-based.

Nevertheless, I completely agree that not all archaeological scientists are familiar with Python, and accessibility is crucial. Therefore, I have made the following changes to try to rectify the challenges you faced:

1) I asked additional people (including non-python users) to test the executable and source code on windows and Linux and let me know any challenges that they faced.

2) I have updated the github tutorial (which is also present in the Zenodo README) to reduce these challenges and increase the clarity and accessibility further.

3) I have refactored the codebase and added many more descriptive comments, to make it more ordered and clearer for users seeking to understand the script itself

If you are keen to use The Demodifier and continue to run into issues- please let me know and I am happy to help!

- line 170: Please spell MaxQuant with Q throughout the document

Thank you, “MaxQuant” is now corrected throughout the text

- line 167-170: Does the user have to create itself a .cvs file with "Sequence" and "Modifications" columns? Why the script does not require unmodified Mascot or MaxQuant output files (such as evidence.txt MQ file), and the script extracts Sequence and Modifications information of interest with a few additional commands? Automating data treatment will facilitate the user experience for non-experts.

Thank you, this is a great point. You are correct that in the submitted version the author had to create the Demodifier input file by copying and pasting the peptide sequences and modifications of interests into a csv and saving it. This was because usually LCA analysis is only done on a subset of peptides from the Mascot or Maxquant output file, for instance after peptides from common contaminant or otherwise irrelevant proteins have been filtered out.

Nevertheless, I completely agree that there are use-cases in which a researcher may wish to apply The Demodifier to all the peptides in their Mascot or Maxquant output file. Therefore, I have updated the script to accept Mascot output csvs and Maxquant evidence files, while maintaining its ability to analyse simple user-made "Sequence" and "Modifications" files.

To reflect this change, I have updated line 175 as follows:

“Briefly, The Demodifier accepts as input a CSV or TSV containing one column called “Sequence” or “pep_seq” containing peptide sequences, and one column called “Modifications” or “pep_var_mod” containing their corresponding variable modifications (if detected) in either Mascot or MaxQuant format. This input file may be either an unaltered Mascot output csv, or MaxQuant evidence.txt file, or a user-created file containing only the relevant columns.”

I have also created a new static Zenodo version of the updated script (v1.6.0) , which is linked via its DOI in the data availability section of the script (https://doi.org/10.5281/zenodo.17689841).

- Fig. 3, box C: The "Q↔E" in Chemistry means a reversible conversion of Q to E. Do you mean "Q/E → pyroGlu" ?

Thank you, good point, that is not quite what I mean. I have changed it to “pyroglutamic acid (Q→E/ E→Q)”, as these are the two substitutions simulated at this point in the script. (i.e. if “Glu->pyro-Glu” was detected by search software , The Demodifier creates permutations based on E being substituted for Q, or if “Gln->pyro-Glu” was detected by search software, Q substituted for E)

- line 393: As you used β for some proteins, please use ɣ-hordein-1 instead.

Thanks for spotting this, I have made this change at line 447.

- line 404: Same with ɑ

Thanks again, I have made this change throughout the text

- line 393: Please add the UniProt code for each protein mentioned.

Thank you for your suggestion. Unfortunately it is not possible to add a single Uniprot code to each protein mentioned in the text, as all peptides mentioned in the text match to multiple distinct Unipro

---

## [Editor Report · Decision Letter 1]

8 Dec 2025

The Demodifier: a tool for screening modification-induced alternate peptide taxonomy in palaeoproteomics

PONE-D-25-50945R1

Dear Dr. Evans,

We’re pleased to inform you that your manuscript has been judged scientifically suitable for publication and will be formally accepted for publication once it meets all outstanding technical requirements.

Kind regards,

Enrico Greco

Academic Editor

PLOS One
---

## [Editor Report · Acceptance letter]

PONE-D-25-50945R1

PLOS One

Dear Dr. Evans,

I'm pleased to inform you that your manuscript has been deemed suitable for publication in PLOS One. Congratulations! Your manuscript is now being handed over to our production team.

Kind regards,

on behalf of

Dr. Enrico Greco

Academic Editor

PLOS One